# EfficientFormer: Vision Transformers at MobileNet Speed

**Yanyu Li**[1,2,†]  **Geng Yuan**[1,2,†]  **Yang Wen**[1]  **Ju Hu**[1]  **Georgios Evangelidis**[1]
**Sergey Tulyakov**[1]  **Yanzhi Wang**[2]  **Jian Ren**[1]
[1]Snap Inc.   [2]Northeastern University

## Abstract

Vision Transformers (ViT) have shown rapid progress in computer vision tasks, achieving promising results on various benchmarks. However, due to the massive number of parameters and model design, *e.g.*, attention mechanism, ViT-based models are generally times slower than lightweight convolutional networks. Therefore, the deployment of ViT for real-time applications is particularly challenging, especially on resource-constrained hardware such as mobile devices. Recent efforts try to reduce the computation complexity of ViT through network architecture search or hybrid design with MobileNet block, yet the inference speed is still unsatisfactory. This leads to an important question: *can transformers run as fast as MobileNet while obtaining high performance?* To answer this, we first revisit the network architecture and operators used in ViT-based models and identify inefficient designs. Then we introduce a dimension-consistent pure transformer (without MobileNet blocks) as a design paradigm. Finally, we perform latency-driven slimming to get a series of final models dubbed EfficientFormer. Extensive experiments show the superiority of EfficientFormer in performance and speed on mobile devices. Our fastest model, EfficientFormer-L1, achieves 79.2% top-1 accuracy on ImageNet-1K with only 1.6 ms inference latency on iPhone 12 (compiled with CoreML), which runs as fast as MobileNetV2×1.4 (1.6 ms, 74.7% top-1), and our largest model, EfficientFormer-L7, obtains 83.3% accuracy with only 7.0 ms latency. Our work proves that properly designed transformers can reach *extremely low latency* on mobile devices while maintaining *high performance*[1].

## 1   Introduction

The transformer architecture [1], initially designed for Natural Language Processing (NLP) tasks, introduces the Multi-Head Self Attention (MHSA) mechanism that allows the network to model long-term dependencies and is easy to parallelize. In this context, Dosovitskiy *et al.* [2] adapt the attention mechanism to 2D images and propose Vision Transformer (ViT): the input image is divided into non-overlapping patches, and the inter-patch representations are learned through MHSA without inductive bias. ViTs demonstrate promising results compared to convolutional neural networks (CNNs) on computer vision tasks. Following this success, several efforts explore the potential of ViT by improving training strategies [3, 4, 5], introducing architecture changes [6, 7], redesigning attention mechanisms [8, 9], and elevating the performance of various vision tasks such as classification [10, 11, 12], segmentation [13, 14], and detection [15, 16].

On the downside, transformer models are usually times slower than competitive CNNs [17, 18]. There are many factors that limit the inference speed of ViT, including the massive number of

---

[†]These authors contributed equally.
[1]Code and models are available at https://github.com/snap-research/EfficientFormer.

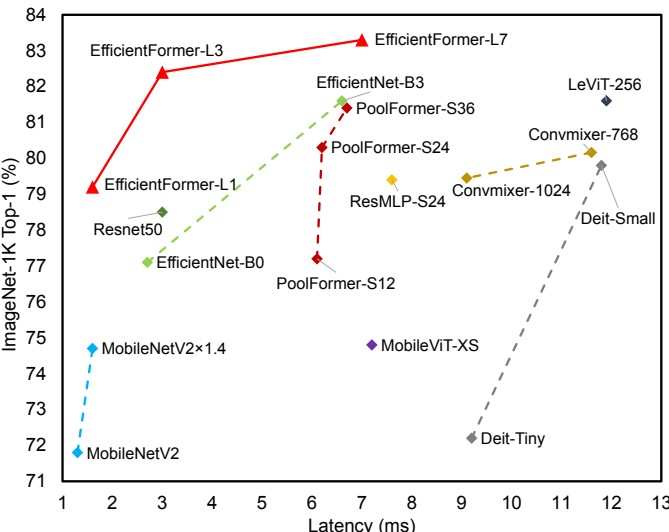

Figure 1: **Inference Speed *vs.* Accuracy.** All models are trained on ImageNet-1K and measured by iPhone 12 with CoreMLTools to get latency. Compared to CNNs, EfficientFormer-L1 runs $40\%$ faster than EfficientNet-B0, while achieves $2.1\%$ higher accuracy. For the latest MobileViT-XS, EfficientFormer-L7 runs $0.2$ ms faster with $8.5\%$ higher accuracy.

parameters, quadratic-increasing computation complexity with respect to token length, non-foldable normalization layers, and lack of compiler level optimizations (*e.g.*, Winograd for CNN [19]). The high latency makes transformers impractical for real-world applications on resource-constrained hardware, such as augmented or virtual reality applications on mobile devices and wearables. As a result, lightweight CNNs [20, 21, 22] remain the default choice for real-time inference.

To alleviate the latency bottleneck of transformers, many approaches have been proposed. For instance, some efforts consider designing new architectures or operations by changing the linear layers with convolutional layers (CONV) [23], combining self-attention with MobileNet blocks [24], or introducing sparse attention [25, 26, 27], to reduce the computational cost, while other efforts leverage network searching algorithm [28] or pruning [29] to improve efficiency. Although the computation-performance trade-off has been improved by existing works, the fundamental question that relates to the applicability of transformer models remains unanswered: *Can powerful vision transformers run at MobileNet speed and become a default option for edge applications?* This work provides a study towards the answer through the following contributions:

- First, we revisit the design principles of ViT and its variants through latency analysis (Sec. 3). Following existing work [18], we utilize iPhone 12 as the testbed and publicly available CoreML [30] as the compiler, since the mobile device is widely used and the results can be easily reproduced.

- Second, based on our analysis, we identify inefficient designs and operators in ViT and propose a new dimension-consistent design paradigm for vision transformers (Sec. 4.1).

- Third, starting from a supernet with the new design paradigm, we propose a simple yet effective latency-driven slimming method to obtain a new family of models, namely, EfficientFormers (Sec. 4.2). We directly optimize for inference speed instead of MACs or number of parameters [31, 32, 33].

Our fastest model, EfficientFormer-L1, achieves $79.2\%$ top-1 accuracy on ImageNet-1K [34] classification task with only $1.6$ ms inference time (averaged over $1,000$ runs), which runs as fast as MobileNetV2×1.4 and wields $4.5\%$ *higher* top-1 accuracy (more results in Fig. 1 and Tab. 1). The promising results demonstrate that latency is no longer an obstacle for the widespread adoption of vision transformers. Our largest model, EfficientFormer-L7, achieves $83.3\%$ accuracy with only $7.0$ ms latency, outperforms ViT×MobileNet hybrid designs (MobileViT-XS, $74.8\%$, $7.2$ms) by a large margin. Additionally, we observe superior performance by employing EfficientFormer as

the backbone in image detection and segmentation benchmarks (Tab. 2). We provide a preliminary answer to the aforementioned question, *ViTs can achieve ultra fast inference speed and wield powerful performance at the same time.* We hope our EfficientFormer can serve as a strong baseline and inspire followup works on the edge deployment of vision transformers.

## 2 Related Work

Transformers are initially proposed to handle the learning of long sequences in NLP tasks [1]. Dosovitskiy *et al.* [2] and Carion *et al.* [15] adapt the transformer architecture to classification and detection, respectively, and achieve competitive performance against CNN counterparts with stronger training techniques and larger-scale datasets. DeiT [3] further improves the training pipeline with the aid of distillation, eliminating the need for large-scale pretraining [35]. Inspired by the competitive performance and global receptive field of transformer models, follow-up works are proposed to refine the architecture [36, 37], explore the relationship between CONV nets and ViT [38, 39, 40], and adapt ViT to different computer vision tasks [13, 41, 42, 43, 44, 45, 46]. Other research efforts explore the essence of attention mechanism and propose insightful variants of token mixer, *e.g.*, local attention [8], spatial MLP [47, 48], and pooling-mixer [6].

Despite the success in most vision tasks, ViT-based models cannot compete with the well-studied lightweight CNNs [21, 49] when the inference speed is the major concern [50, 51, 52], especially on resource-constrained edge devices [17]. To accelerate ViT, many approaches have been introduced with different methodologies, such as proposing new architectures or modules [53, 54, 55, 56, 57, 58], re-thinking self-attention and sparse-attention mechanisms [59, 60, 61, 62, 63, 64, 65], and utilizing search algorithms that are widely explored in CNNs to find smaller and faster ViTs [66, 28, 29, 67]. Recently, LeViT [23] proposes a CONV-clothing design to accelerate vision transformer. However, in order to perform MHSA, the 4D features need to be frequently reshaped into flat patches, which is still expensive to compute on edge resources (Fig. 2). Likewise, MobileViT [18] introduces a hybrid architecture that combines lightweight MobileNet blocks (with point-wise and depth-wise CONV) and MHSA blocks; the former is placed at early stages in the network pipeline to extract low-level features, while the latter is placed in late stages to enjoy the global receptive field. Similar approach has been explored by several works [24, 28] as a straightforward strategy to reduce computation.

Different from existing works, we aim at pushing the latency-performance boundary of pure vision transformers instead of relying on hybrid designs, and directly optimize for mobile latency. Through our detailed analysis (Sec. 3), we propose a new design paradigm (Sec. 4.1), which can be further elevated through architecture search (Sec. 4.2).

## 3 On-Device Latency Analysis of Vision Transformers

Most existing approaches optimize the inference speed of transformers through computation complexity (MACs) or throughput (images/sec) obtained from server GPU [23, 28]. While such metrics do not reflect the real on-device latency. To have a clear understanding of which operations and design choices slow down the inference of ViTs on edge devices, we perform a comprehensive latency analysis over a number of models and operations, as shown in Fig. 2, whereby the following observations are drawn.

**Observation 1**: *Patch embedding with large kernel and stride is a speed bottleneck on mobile devices.*

Patch embedding is often implemented with a non-overlapping convolution layer that has large kernel size and stride [3, 55]. A common belief is that the computation cost of the patch embedding layer in a transformer network is unremarkable or negligible [2, 6]. However, our comparison in Fig. 2 between models with large kernel and stride for patch embedding, *i.e.*, DeiT-S [3] and PoolFormer-S24 [6], and the models without it, *i.e.*, LeViT-256 [23] and EfficientFormer, shows that patch embedding is instead a speed bottleneck on mobile devices.

Large-kernel convolutions are not well supported by most compilers and cannot be accelerated through existing algorithms like Winograd [19]. Alternatively, the non-overlapping patch embedding can be replaced by a convolution stem with fast downsampling [68, 69, 23] that consists of several hardware-efficient $3 \times 3$ convolutions (Fig. 3).

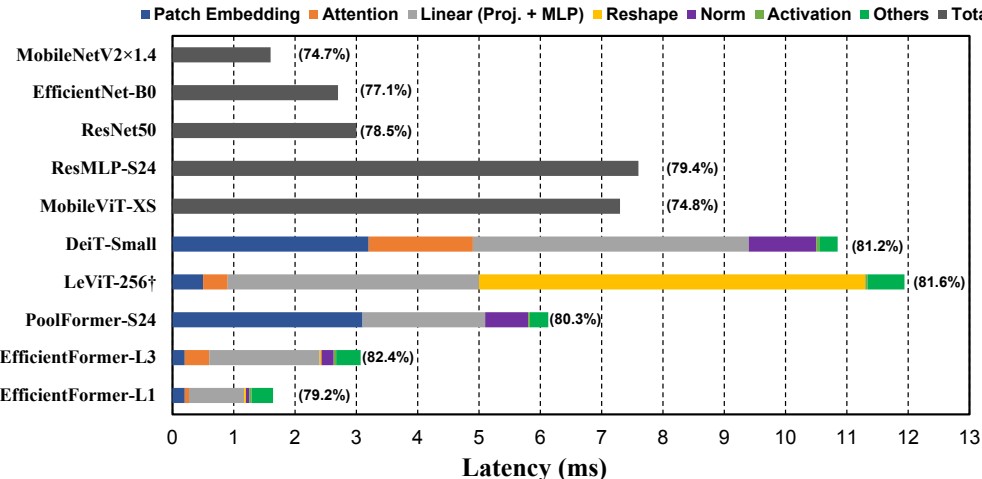

Figure 2: **Latency profiling.** Results are obtained on iPhone 12 with CoreML. The on-device speed for CNN (MobileNetV2×1.4, ResNet50, and EfficientNet-B0), ViT-based models (DeiT-Small, LeViT-256, PoolFormer-S24, and EfficientFormer), and various operators are reported. The latency of models and operations are denoted with different color. (·) is the top-1 accuracy on ImageNet-1K. †LeViT uses HardSwish which is not well supported by CoreML, we replace it with GeLU for fair comparison.

**Observation 2**: *Consistent feature dimension is important for the choice of token mixer. MHSA is not necessarily a speed bottleneck.*

Recent work extends ViT-based models to the MetaFormer architecture [6] consisting of MLP blocks and unspecified token mixers. Selecting a token mixer is an essential design choice when building ViT-based models. The options are many—the conventional MHSA mixer with a global receptive field, more sophisticated shifted window attention [8], or a non-parametric operator like pooling [6].

We narrow the comparison to the two token mixers, pooling and MHSA, where we choose the former for its simplicity and efficiency, while the latter for better performance. More complicated token mixers like shifted window [8] are currently not supported by most public mobile compilers and we leave them outside our scope. Furthermore, we do not use depth-wise convolution to replace pooling [70] as we focus on building architecture without the aid of lightweight convolutions.

To understand the latency of the two token mixers, we perform the following two comparisons:

- First, by comparing PoolFormer-s24 [6] and LeViT-256 [23], we observe that the `Reshape` operation is a bottleneck for LeViT-256. The majority of LeViT-256 is implemented with CONV on 4D tensor, requiring frequent reshaping operations when forwarding features into MHSA since the attention has to be performed on patchified 3D tensor (discarding the extra dimension of attention heads). The extensive usage of `Reshape` limits the speed of LeViT on mobile devices (Fig. 2). On the other hand, pooling naturally suits the 4D tensor when the network primarily consists of CONV-based implementations, *e.g.*, CONV $1 \times 1$ as MLP implementation and CONV stem for downsampling. As a result, PoolFormer exhibits faster inference speed.

- Second, by comparing DeiT-Small [3] and LeViT-256 [23], we find that MHSA does not bring significant overhead on mobiles if the feature dimensions are consistent and `Reshape` is not required. Though much more computation intensive, DeiT-Small with a consistent 3D feature can achieve comparable speed to the new ViT variant, *i.e.*, LeViT-256.

In this work, we propose a dimension-consistent network (Sec. 4.1) with both 4D feature implementation and 3D MHSA, but the inefficient frequent `Reshape` operations are eliminated.

**Observation 3**: *CONV-BN is more latency-favorable than LN (GN)-Linear and the accuracy drawback is generally acceptable.*

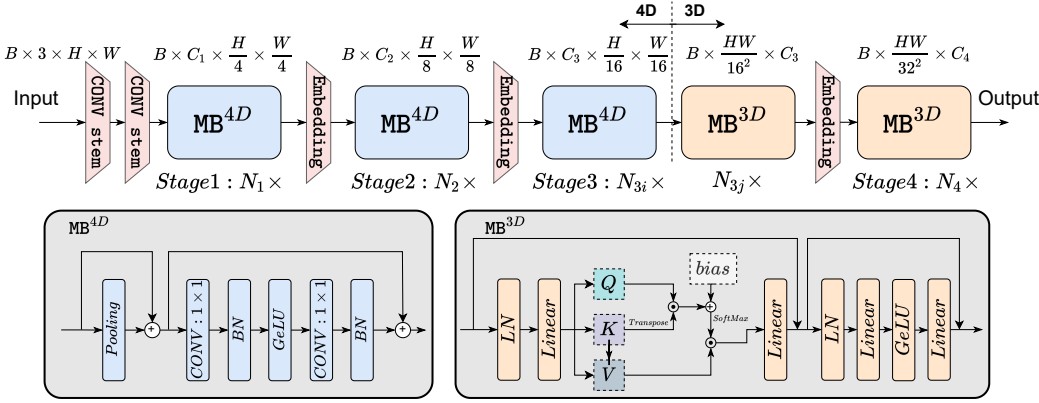

Figure 3: **Overview of EfficientFormer.** The network starts with a convolution stem as patch embedding, followed by MetaBlock (MB). The $MB^{4D}$ and $MB^{3D}$ contain different token mixer configurations, *i.e.*, local pooling or global multi-head self-attention, arranged in a dimension-consistent manner.

Choosing the MLP implementation is another essential design choice. Usually, one of the two options is selected: layer normalization (LN) with 3D linear projection (proj.) and CONV $1 \times 1$ with batch normalization (BN). CONV-BN is more latency favorable because BN can be folded into the preceding convolution for inference speedup, while dynamic normalizations, such as LN and GN, still collects running statistics at the inference phase, thus contributing to latency. From the analysis of DeiT-Small and PoolFormer-S24 in Fig. 2 and previous work [17], the latency introduced by LN constitutes around $10\% - 20\%$ latency of the whole network.

Based on our ablation study in Appendix Tab. 3, CONV-BN only slightly downgrades performance compared to GN and achieves comparable results to channel-wise LN. In this work, we apply CONV-BN as much as possible (in all latent 4D features) for the latency gain with a negligible performance drop, while using LN for the 3D features, which aligns with the original MHSA design in ViT and yields better accuracy.

**Observation 4**: *The latency of nonlinearity is hardware and compiler dependent.*

Lastly, we study nonlinearity, including GeLU, ReLU, and HardSwish. Previous work [17] suggests GeLU is not efficient on hardware and slows down inference. However, we observe GeLU is well supported by iPhone 12 and hardly slower than its counterpart, ReLU. On the contrary, HardSwish is surprisingly slow in our experiments and may not be well supported by the compiler (LeViT-256 latency with HardSwish is $44.5$ ms while with GeLU $11.9$ ms). We conclude that nonlinearity should be determined on a case-by-case basis given specific hardware and compiler at hand. We believe that most of the activations will be supported in the future. In this work, we employ GeLU activations.

## 4 Design of EfficientFormer

Based on the latency analysis, we propose the design of EfficientFormer, demonstrated in Fig. 3. The network consists of a patch embedding (`PatchEmbed`) and stack of meta transformer blocks, denoted as `MB`:

$$\mathcal{Y} = \prod_{i}^{m} \texttt{MB}_i(\texttt{PatchEmbed}(\mathcal{X}_0^{B,3,H,W})), \tag{1}$$

where $\mathcal{X}_0$ is the input image with batch size as $B$ and spatial size as $[H, W]$, $\mathcal{Y}$ is the desired output, and $m$ is the total number of blocks (depth). MB consists of unspecified token mixer (`TokenMixer`) followed by a MLP block and can be expressed as follows:

$$\mathcal{X}_{i+1} = \texttt{MB}_i(\mathcal{X}_i) = \texttt{MLP}(\texttt{TokenMixer}(\mathcal{X}_i)), \tag{2}$$

where $\mathcal{X}_{i|i>0}$ is the intermediate feature that forwarded into the $i^{th}$ MB. We further define Stage (or S) as the stack of several MetaBlocks that processes the features with the same spatial size, such as $N_1 \times$ in Fig. 3 denoting $S_1$ has $N_1$ MetaBlocks. The network includes 4 Stages. Among each Stage, there

is an embedding operation to project embedding dimension and downsample token length, denoted as `Embedding` in Fig. 3. With the above architecture, EfficientFormer is a fully transformer-based model without integrating MobileNet structures. Next, we dive into the details of the network design, specifically, the architecture details and the search algorithm.

## 4.1 Dimension-Consistent Design

With the observations in Sec. 3, we propose a dimension consistent design which splits the network into a 4D partition where operators are implemented in CONV-net style ($\texttt{MB}^{4D}$), and a 3D partition where linear projections and attentions are performed over 3D tensor to enjoy the global modeling power of MHSA without sacrificing efficiency ($\texttt{MB}^{3D}$), as shown in Fig. 3. Specifically, the network starts with 4D partition, while 3D partition is applied in the last stages. Note that Fig. 3 is just an instance, the actual length of 4D and 3D partition is specified later through architecture search.

First, input images are processed by a CONV stem with two $3 \times 3$ convolutions with stride 2 as patch embedding,

$$\mathcal{X}_1^{B,C_{j|j=1},\frac{H}{4},\frac{W}{4}} = \texttt{PatchEmbed}(\mathcal{X}_0^{B,3,H,W}), \tag{3}$$

where $C_j$ is the channel number (width) of the $j$th stage. Then the network starts with $\texttt{MB}^{4D}$ with a simple `Pool` mixer to extract low level features,

$$
\begin{aligned}
\mathcal{I}_i &= \texttt{Pool}(\mathcal{X}_i^{B,C_j,\frac{H}{2^{j+1}},\frac{W}{2^{j+1}}}) + \mathcal{X}_i^{B,C_j,\frac{H}{2^{j+1}},\frac{W}{2^{j+1}}}, \\
\mathcal{X}_{i+1}^{B,C_j,\frac{H}{2^{j+1}},\frac{W}{2^{j+1}}} &= \texttt{Conv}_B(\texttt{Conv}_{B,G}(\mathcal{I}_i)) + \mathcal{I}_i,
\end{aligned}
\tag{4}
$$

where $\texttt{Conv}_{B,G}$ refers to whether the convolution is followed by BN and GeLU, respectively. Note here we do not employ Group or Layer Normalization (LN) before the `Pool` mixer as in [6], since the 4D partition is CONV-BN based design, thus there exists a BN in front of each `Pool` mixer.

After processing all the $\texttt{MB}^{4D}$ blocks, we perform a one-time reshaping to transform the features size and enter 3D partition. $\texttt{MB}^{3D}$ follows conventional ViT structure, as in Fig. 3. Formally,

$$
\begin{aligned}
\mathcal{I}_i &= \texttt{Linear}(\texttt{MHSA}(\texttt{Linear}(\texttt{LN}(\mathcal{X}_i^{B,\frac{HW}{4^{j+1}},C_j})))) + \mathcal{X}_i^{B,\frac{HW}{4^{j+1}},C_j}, \\
\mathcal{X}_{i+1}^{B,\frac{HW}{4^{j+1}},C_j} &= \texttt{Linear}(\texttt{Linear}_G(\texttt{LN}(\mathcal{I}_i))) + \mathcal{I}_i,
\end{aligned}
\tag{5}
$$

where $\texttt{Linear}_G$ denotes the `Linear` followed by GeLU, and

$$\texttt{MHSA}(Q,K,V) = \texttt{Softmax}(\frac{Q \cdot K^T}{\sqrt{C_j}} + b) \cdot V, \tag{6}$$

where $Q, K, V$ represents query, key, and values learned by the linear projection, and $b$ is parameterized attention bias as position encodings.

## 4.2 Latency Driven Slimming

**Design of Supernet.** Based on the dimension-consistent design, we build a supernet for searching efficient models of the network architecture shown in Fig. 3 (Fig. 3 shows an example of searched final network). In order to represent such a supernet, we define the MetaPath (MP), which is the collection of possible blocks:

$$
\begin{aligned}
\texttt{MP}_{i,j=1,2} &\in \{\texttt{MB}_i^{4D}, I_i\}, \\
\texttt{MP}_{i,j=3,4} &\in \{\texttt{MB}_i^{4D}, \texttt{MB}_i^{3D}, I_i\},
\end{aligned}
\tag{7}
$$

where $I$ represents identity path, $j$ denotes the $j^{th}$ Stage, and $i$ denotes the $i^{th}$ block. The supernet can be illustrated by replacing `MB` in Fig. 3 with `MP`.

As in Eqn. 7, in $\texttt{S}_1$ and $\texttt{S}_2$ of the supernet, each block can select from $\texttt{MB}^{4D}$ or $I$, while in $\texttt{S}_3$ and $\texttt{S}_4$, the block can be $\texttt{MB}^{3D}$, $\texttt{MB}^{4D}$, or $I$. We only enable $\texttt{MB}^{3D}$ in the last two Stages for two reasons. First, since the computation of MHSA grows quadratically with respect to token length, integrating it in early Stages would largely increase the computation cost. Second, applying the global MHSA to the last Stages aligns with the intuition that early stages in the networks capture low-level features, while late layers learn long-term dependencies.

**Searching Space.** Our searching space includes $C_j$ (the width of each Stage), $N_j$ (the number of blocks in each Stage, *i.e.*, depth), and last $\mathbb{N}$ blocks to apply $\texttt{MB}^{3D}$.

**Searching Algorithm.** Previous hardware-aware network searching methods generally rely on hardware deployment of each candidate in search space to obtain the latency, which is time consuming [71]. In this work, we propose a simple, fast yet effective gradient-based search algorithm to obtain a candidate network that just needs to train the supernet for once. The algorithm has three major steps.

First, we train the supernet with Gumbel Softmax sampling [72] to get the importance score for the blocks within each $\texttt{MP}$, which can be expressed as

$$\mathcal{X}_{i+1} = \sum_n \frac{e^{(\alpha_i^n + \epsilon_i^n)/\tau}}{\sum_n e^{(\alpha_i^n + \epsilon_i^n)/\tau}} \cdot \texttt{MP}_{i,j}(\mathcal{X}_i), \tag{8}$$

where $\alpha$ evaluates the importance of each block in $\texttt{MP}$ as it represents the probability to select a block, *e.g.*, $\texttt{MB}^{4D}$ or $\texttt{MB}^{3D}$ for the $i^{th}$ block. $\epsilon \sim U(0,1)$ ensures exploration, $\tau$ is the temperature, and $n$ represents the type of blocks in $\texttt{MP}$, *i.e.*, $n \in \{4D, I\}$ for $\texttt{S}_1$ and $\texttt{S}_2$, and $n \in \{4D, 3D, I\}$ for $\texttt{S}_3$ and $\texttt{S}_4$. By using Eqn. 8, the derivatives with respect to network weights and $\alpha$ can be computed easily. The training follows the standard recipe (see Sec. 5.1) to obtain the trained weights and architecture parameter $\alpha$.

Second, we build a latency lookup table by collecting the on-device latency of $\texttt{MB}^{4D}$ and $\texttt{MB}^{3D}$ with different widths (multiples of 16).

Finally, we perform network slimming on the supernet obtained from the first step through latency evaluation using the lookup table. Note that a typical gradient-based searching algorithm simply select the block with largest $\alpha$ [72], which does not fit our scope as it cannot search the width $C_j$. In fact, constructing a multiple-width supernet is memory-consuming and even unrealistic given that each $\texttt{MP}$ has several branches in our design. Instead of directly searching on the complex searching space, we perform a gradual slimming on the single-width supernet as follows.

We first define the importance score for $\texttt{MP}_i$ as $\frac{\alpha_i^{4D}}{\alpha_i^I}$ and $\frac{\alpha_i^{3D} + \alpha_i^{4D}}{\alpha_i^I}$ for $\texttt{S}_{1,2}$ and $\texttt{S}_{3,4}$, respectively. Similarly, the importance score for each Stage can be obtained by summing up the scores for all $\texttt{MP}$ within the Stage. With the importance score, we define the action space that includes three options: 1) select $I$ for the least important $\texttt{MP}$, 2) remove the first $\texttt{MB}^{3D}$, and 3) reduce the width of the least important Stage (by multiples of 16). Then, we calculate the resulting latency of each action through lookup table, and evaluate the accuracy drop of each action. Lastly, we choose the action based on *per-latency accuracy drop ($\frac{-\%}{ms}$)*. This process is performed iteratively until target latency is achieved. We show more details of the algorithm in Appendix.

## 5 Experiments and Discussion

We implement EfficientFormer through PyTorch 1.11 [73] and Timm library [74], which is the common practice in recent arts [18, 6]. Our models are trained on a cluster with NVIDIA A100 and V100 GPUs. The inference speed on iPhone 12 (A14 bionic chip) is measured with iOS version 15 and averaged over $1,000$ runs, with all available computing resources (NPU), or CPU only. CoreMLTools is used to deploy the run-time model. In addition, we provide latency analysis on Nvidia A100 GPU with batch size 64 to exploit hardware roofline. The trained PyTorch models are deployed in ONNX format and are compiled with TensorRT. We report GPU runtime that excludes preprocessing. We provide the detailed network architecture and more ablation studies in Appendix 6.

### 5.1 Image Classification

All EfficientFormer models are trained from scratch on ImageNet-1K dataset [34] to perform the image classification task. We employ standard image size ($224 \times 224$) for both training and testing. We follow the training recipe from DeiT [3] but mainly report results with 300 training epochs to have the comparison with other ViT-based models. We use AdamW optimizer [75, 76], warm-up training with 5 epochs, and a cosine annealing learning rate schedule. The initial learning rate is set as $10^{-3} \times (batch\ size/1024)$ and the minimum learning rate is $10^{-5}$. The teacher model for distillation

Table 1: **Comparison results on ImgeNet-1K.** The latency results are tested on iPhone Neural Engine (NPU), iPhone CPU and Nvidia A100 GPU correspondingly. Note that for mobile speed, we report latency per frame, while on A100 GPU, we report latency per batch size 64 to maximum resource utilization. Hybrid refers to a mixture of MobileNet blocks and ViT blocks. (-) refers to unrevealed or unsupported models. †Latency measured with GeLU activation for fair comparison, the original LeViT-256 model with HardSwish activations runs at $44.5$ ms. Different training seeds lead to less than $\pm 0.2\%$ fluctuation in accuracy for EfficientFormer, and the error for latency benchmark is less than $\pm 0.1$ ms.

| Model | Type | Params(M) | GMACs | Train. Epoch | Top-1(%) | Latency (ms) | | |
| --- | --- | --- | --- | --- | --- | --- | --- | --- |
| | | | | | | NPU | CPU | A100 |
| MobileNetV2×1.0 | CONV | 3.5 | 0.3 | 300 | 71.8 | 1.3 | 8.0 | 5.0 |
| MobileNetV2×1.4 | CONV | 6.1 | 0.6 | 300 | 74.7 | 1.6 | 10.7 | 7.3 |
| ResNet50 | CONV | 25.5 | 4.1 | 300 | 78.5 | 3.0 | 29.4 | 9.0 |
| EfficientNet-B0 | CONV | 5.3 | 0.4 | 350 | 77.1 | 2.7 | 14.5 | 10.0 |
| EfficientNet-B3 | CONV | 12.0 | 1.8 | 350 | 81.6 | 6.6 | 52.6 | 35.0 |
| EfficientNet-B5 | CONV | 30.0 | 9.9 | 350 | 83.6 | 23.0 | 258.8 | 141.0 |
| DeiT-T | Attention | 5.9 | 1.2 | 300/1000 | 74.5/76.6 | 9.2 | 16.7 | 7.1 |
| DeiT-S | Attention | 22.5 | 4.5 | 300/1000 | 81.2/82.6 | 11.8 | 41.0 | 15.5 |
| PVT-Small | Attention | 24.5 | 3.8 | 300 | 79.8 | 24.4 | 89.5 | 23.8 |
| T2T-ViT-14 | Attention | 21.5 | 4.8 | 310 | 81.5 | - | - | 21.0 |
| Swin-Tiny | Attention | 29 | 4.5 | 300 | 81.3 | - | - | 22.0 |
| CSwin-T | Attention | 23 | 4.3 | 300 | 82.7 | - | - | 28.7 |
| PoolFormer-s12 | Pool | 12 | 2.0 | 300 | 77.2 | 6.1 | 59.0 | 14.5 |
| PoolFormer-s24 | Pool | 21 | 3.6 | 300 | 80.3 | 6.2 | 126.7 | 28.2 |
| PoolFormer-s36 | Pool | 31 | 5.2 | 300 | 81.4 | 6.7 | 192.6 | 41.2 |
| ResMLP-S24 | SMLP | 30 | 6.0 | 300 | 79.4 | 7.6 | 40.2 | 17.4 |
| Convmixer-768 | Hybrid | 21.1 | 20.7 | 300 | 80.2 | 11.6 | 29.3 | - |
| LeViT-256 | Hybrid | 18.9 | 1.1 | 1000 | 81.6 | 11.9 † | 13.5 | 4.5 |
| NASViT-A5 | Hybrid | - | 0.76 | 360 | 81.8 | - | - | - |
| MobileViT-XS | Hybrid | 2.3 | 0.7 | 300 | 74.8 | 7.2 | 26.5 | 11.7 |
| MobileFormer-508M | Hybrid | 14.0 | 0.51 | 450 | 79.3 | 13.2 | 22.2 | 14.6 |
| EfficientFormer-L1 | MetaBlock | 12.3 | 1.3 | 300/1000 | **79.2/80.2** | **1.6** | **11.5** | **6.2** |
| EfficientFormer-L3 | MetaBlock | 31.3 | 3.9 | 300 | **82.4** | **3.0** | **28.2** | **13.9** |
| EfficientFormer-L7 | MetaBlock | 82.1 | 10.2 | 300 | **83.3** | **7.0** | **67.7** | **30.7** |

is RegNetY-16GF [77] pretrained on ImageNet with $82.9\%$ top-1 accuracy. Results are demonstrated in Tab. 1 and Fig. 1

**Comparison to CNNs.** Compared with the widely used CNN-based models, EfficientFormer achieves a better trade-off between accuracy and latency. On iPhone Neural Engine, EfficientFormer-L1 runs at MobileNetV2×1.4 speed while achieving $4.5\%$ higher top-1 accuracy. In addition, EfficientFormer-L3 runs at a similar speed to EfficientNet-B0 while achieving relative $5.3\%$ higher top-1 accuracy. For the models with high performance ($> 83\%$ top-1), EfficientFormer-L7 runs more than 3× faster than EfficientNet-B5, demonstrating the advantageous performance of our models. Moreover on desktop GPU (A100), EfficientFormer-L1 runs $38\%$ faster than EfficientNet-B0 while achieving $2.1\%$ higher top-1 accuracy. EfficientFormer-L7 runs 4.6× faster than EfficientNet-B5. These results allow us to answer the central question raised earlier; *ViTs do not need to sacrifice latency to achieve good performance, and an accurate ViT can still have ultra-fast inference speed as lightweight CNNs do.*

**Comparison to ViTs.** Conventional ViTs are still under-performing CNNs in terms of latency. For instance, DeiT-Tiny achieves similar accuracy to EfficientNet-B0 while it runs 3.4× slower. However, EfficientFormer performs like other transformer models while running times faster. EfficientFormer-L3 achieves higher accuracy than DeiT-Small ($82.4\%$ *vs.* $81.2\%$) while being 4× faster. It is notable that though the recent transformer variant, PoolFormer [6], naturally has a consistent 4D architecture and runs faster compared to typical ViTs, the absence of global MHSA greatly limits the performance upper-bound. EfficientFormer-L3 achieves $1\%$ higher top-1 accuracy than PoolFormer-S36, while being 3× faster on Nvidia A100 GPU, 2.2× faster on iPhone NPU and 6.8× faster on iPhone CPU.

**Comparison to Hybrid Designs.** Existing hybrid designs, *e.g.*, LeViT-256 and MobileViT, still struggle with the latency bottleneck of ViTs and can hardly outperform lightweight CNNs. For example, LeViT-256 runs slower than DeiT-Small while having $1\%$ lower top-1 accuracy. For MobileViT, which is a hybrid model with both MHSA and MobileNet blocks, we observe that it

Table 2: **Comparison results using EfficientFormer as backbone.** Results on object detection & instance segmentation are obtained from COCO 2017. Results on semantic segmentation are obtained from ADE20K.

| Backbone | Detection & Instance Segmentation | | | | | | Semantic |
|---|---|---|---|---|---|---|---|
| | $\mathbf{AP}^{box}$ | $\mathbf{AP}^{box}_{50}$ | $\mathbf{AP}^{box}_{75}$ | $\mathbf{AP}^{mask}$ | $\mathbf{AP}^{mask}_{50}$ | $\mathbf{AP}^{mask}_{75}$ | mIoU(%) |
| ResNet18 | 34.0 | 54.0 | 36.7 | 31.2 | 51.0 | 32.7 | 32.9 |
| PoolFormer-S12 | 37.3 | 59.0 | 40.1 | 34.6 | 55.8 | 36.9 | 37.2 |
| EfficientFormer-L1 | **37.9** | **60.3** | **41.0** | **35.4** | **57.3** | **37.3** | **38.9** |
| ResNet50 | 38.0 | 58.6 | 41.4 | 34.4 | 55.1 | 36.7 | 36.7 |
| PoolFormer-S24 | 40.1 | 62.2 | 43.4 | 37.0 | 59.1 | 39.6 | 40.3 |
| EfficientFormer-L3 | **41.4** | **63.9** | **44.7** | **38.1** | **61.0** | **40.4** | **43.5** |
| ResNet101 | 40.4 | 61.1 | 44.2 | 36.4 | 57.7 | 38.8 | 38.8 |
| PoolFormer-S36 | 41.0 | 63.1 | 44.8 | 37.7 | 60.1 | 40.0 | 42.0 |
| EfficientFormer-L7 | **42.6** | **65.1** | **46.1** | **39.0** | **62.2** | **41.7** | **45.1** |

is significantly slower than CNN counterparts, *e.g.*, MobileNetV2 and EfficientNet-B0, while the accuracy is not satisfactory either (2.3% lower than EfficientNet-B0). Thus, simply trading-off MHSA with MobileNet blocks can hardly push forward the Pareto curve, as in Fig. 1. In contrast, EfficientFormer, as pure transformer-based model, can maintain high performance while achieving ultra-fast inference speed. EfficientFormer-L1 has 4.4% higher top-1 accuracy than MobileViT-XS and runs much faster across different hardware and compilers (1.9× faster on Nvidia A100 GPU Computing, 2.3× faster on iPhone CPU, and 4.5× faster on iPhone NPU). At a similar inference time, EfficientFormer-L7 outperforms MobileViT-XS by $8.5\%$ top-1 accuracy on ImageNet, demonstrating the superiority of our design.

## 5.2 EfficientFormer as Backbone

**Object Detection and Instance Segmentation.** We follow the implementation of Mask-RCNN [78] to integrate EfficientFormer as the backbone and verify performance. We experiment over COCO-2017 [79] which contains training and validations sets of 118K and 5K images, respectively. The EfficientFormer backbone is initialized with ImageNet-1K pretrained weights. Similar to prior work [6], we use AdamW optimizer [75, 76] with initial learning rate of $2 \times 10^{-4}$, and train the model for 12 epochs. We set the input size as $1333 \times 800$.

The results for detection and instance segmentation are shown in Tab. 2. EfficientFormers consistently outperform CNN (ResNet) and transformer (PoolFormer) backbones. With similar computation cost, EfficientFormer-L3 outperforms ResNet50 backbone by 3.4 box **AP** and 3.7 mask **AP**, and outperforms PoolFormer-S24 backbone with 1.3 box **AP** and 1.1 mask **AP**, proving that EfficientFormer generalizes well as a strong backbone in vision tasks.

**Semantic Segmentation.** We further validate the performance of EfficientFormer on the semantic segmentation task. We use the challenging scene parsing dataset, ADE20K [80, 81], which contains 20K training images and 2K validation ones covering 150 class categories. Similar to existing work [6], we build EfficientFormer as backbone along with Semantic FPN [82] as segmentation decoder for fair comparison. The backbone is initialized with pretrained weights on ImageNet-1K and the model is trained for 40K iterations with a total batch size of 32 over 8 GPUs. We follow the common practice in segmentation [6, 13], use AdamW optimizer [75, 76], and apply a poly learning rate schedule with power 0.9, starting from a initial learning rate $2 \times 10^{-4}$. We resize and crop input images to $512 \times 512$ for training and shorter side as $512$ for testing (on validation set).

As shown in Tab. 2, EfficientFormer consistently outperforms CNN- and transformer-based backbones by a large margin under a similar computation budget. For example, EfficientFormer-L3 outperforms PoolFormer-S24 by 3.2 mIoU. We show that with global attention, EfficientFormer learns better long-term dependencies, which is beneficial in high-resolution dense prediction tasks.

## 5.3 Discussion

**Relations to MetaFormer.** The design of EfficientFormer is partly inspired by the MetaFormer concept [6]. Compared to PoolFormer, EfficientFormer addresses the dimension mismatch problem,

which is a root cause of inefficient edge inference, thus being capable of utilizing global MHSA without sacrificing speed. Consequently, EfficientFormer exhibits advantageous accuracy performance over PoolFormer. In spite of its fully 4D design, PoolFormer employs inefficient patch embedding and group normalization (Fig. 2), leading to increased latency. Instead, our redesigned 4D partition of EfficientFormer (Fig. 3) is more hardware friendly and exhibits better performance across several tasks.

**Limitations.** (i) Though most designs in EfficientFormer are general-purposed, *e.g.*, dimension-consistent design and 4D block with CONV-BN fusion, the actual speed of EfficientFormer may vary on other platforms. For instance, if GeLU is not well supported while HardSwish is efficiently implemented on specific hardware and compiler, the operator may need to be modified accordingly. (ii) The proposed latency-driven slimming is simple and fast. However, better results may be achieved if search cost is not a concern and an enumeration-based brute search is performed.

# 6 Conclusion

In this work, we show that Vision Transformer can operate at MobileNet speed on mobile devices. Starting from a comprehensive latency analysis, we identify inefficient operators in a series of ViT-based architectures, whereby we draw important observations that guide our new design paradigm. The proposed EfficientFormer complies with a dimension consistent design that smoothly leverages hardware-friendly 4D MetaBlocks and powerful 3D MHSA blocks. We further propose a fast latency-driven slimming method to derive optimized configurations based on our design space. Extensive experiments on image classification, object detection, and segmentation tasks show that Efficient-Former models outperform existing transformer models while being faster than most competitive CNNs. The latency-driven analysis of ViT architecture and the experimental results validate our claim: powerful vision transformers can achieve ultra-fast inference speed on the edge. Future research will further explore the potential of EfficientFormer on several resource-constrained devices.

# Acknowledgment

This work is supported in part by National Science Foundation CCF-1937500.

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
