# Appendix

## A    Latency-Driven Slimming Algorithm

We provide the details of the proposed latency-driven fast slimming in Alg. 1. Formulations of the algorithm can be found in Sec. 4.2. The proposed latency-driven slimming is speed-oriented, which does not require retraining for each sub-network. The importance score for each design choice is estimated based on the trainable architecture parameter $\alpha$.

---

**Algorithm 1** Fast Latency-Driven Slimming based on Importance Estimations

---

**Require:** Latency lookup table $T = \{\text{MB}^{4D}\{dim = 16\times\}, \text{MB}^{3D}\{dim = 16\times\}\}$
**Ensure:** Final latency satisfy budget $\sum T \approx \mathcal{T}$
   *Super-net Pretraining*:
   **for** epoch **do**
      **for** each iter **do**
         **for** $\text{MP}_{i,j}$ **do**
$$\mathcal{X}_{i+1} = \sum_n \frac{e^{(\alpha_i^n + \epsilon_i^n)/\tau}}{\sum_n e^{(\alpha_i^n + \epsilon_i^n)/\tau}} \cdot \text{MP}_{i,j}(\mathcal{X}_i),$$
         **end for**
         $\mathcal{L} \leftarrow criterion(\mathcal{Y}, label)$
         backpropagate $(\mathcal{L})$, update parameters
      **end for**
   **end for**                         ▷ get super-net
   *Latency-driven slimming:*
   Initialize action space A $\in$ {Depth Reduction (DR), Width Reduction (WR), $\text{MB}^{3D}$ Reduction (MR)}
   Compute importance of $\text{MP}_{i,j}$ by $\mathbb{I}_{i,j} = \frac{\alpha_i^{4D}}{\alpha_i^I}, or \frac{\alpha_i^{3D} + \alpha_i^{4D}}{\alpha_i^I}$
   **while** $\sum T > \mathcal{T}$ **do**

   DR $\leftarrow \underset{\mathbb{I}_{i,j}}{\operatorname{argmin}}(\text{MP}_{i,j}), \quad$ WR $\leftarrow \underset{\sum_j \mathbb{I}_{i,j}}{\operatorname{argmin}}(\text{MP}_{i,j}), \quad$ MR $\leftarrow$ First $- \text{MB}^{3D}$,

   Execute Action $= \underset{\frac{\text{accuracy drop}}{T_{i,j}}}{\operatorname{argmin}}(A)$

   **end while**                    ▷ get sub-net with target latency
   *Train the searched architecture from scratch*:
   Similar to super-net training.                ▷ get final model

---

## B    Ablation Analysis

Our major conclusions and speed analysis can be found in Sec. 3 and Fig. 2. Here we include more ablation studies for different design choices, provided in Tab. 3, taking the EfficientFormer-L3 as an example. The latency is measured on iPhone 12 with CoreML, and the top-1 accuracy is obtained from the ImageNet-1K dataset.

**Patch Embedding.** Compared to non-overlap large-kernel patch embedding (V5 in Tab. 3), the proposed convolution stem in EfficientFormer (V1 in Tab. 3) greatly reduces inference latency by $48\%$, while provides $0.7\%$ higher accuracy. We demonstrate that convolution stem [23] is not only beneficial to model convergence and accuracy but also boosts inference speed on the mobile device by a large margin, thus can serve as a good alternative to non-overlapping patch embedding implementations.

**MHSA and Latency-Driven Search.** Without the proposed 3D MHSA and latency-driven search, EfficientFormer downgrades to a pure 4D design with pool mixer, which is similar to PoolFormer [6] (the patch embeddings and normalizations are different). By comparing EfficientFormer with V1 in Tab. 3, we can observe that the integration of 3D MHSA and latency-driven search greatly boost

top-1 accuracy by 2.1% with minimal impact on the inference speed (0.5 ms). The results prove that MHSA with the global receptive field is an essential contribution to model performance. As a result, though enjoying faster inference speed, simply removing MHSA [6] greatly limits the performance upper bound. In EfficientFormer, we smoothly integrate MHSA in a dimension consistent manner, obtaining better performance while simultaneously achieving ultra fast inference speed.

**Normalization.** Apart from the CONV-BN structure in the 4D partition of EfficientFormer, we explore Group Normalization (GN) and channel-wise Layer Normalization (LN) in the 4D partition as employed in the prior work [6]. By comparing V1 and V2 in Tab. 3, we can observe that the GN (V2-GN) can only slightly improve accuracy (0.3% top-1) but incurs latency overhead as it can not be folded at the inference stage. Similarly, applying LN (V2-LN) gets higher latency than BN while the performance improvement is negligible. As a result, we apply the CONV-BN structure in the entire 4D partition in EfficientFormer.

**Activation Functions.** We explore ReLU and HardSwish (V3 and V4 in Tab. 3) in addition to GeLU employed in this work (V1 in Tab. 3). It is widely agreed that ReLU is the simplest and fastest activation function, while GeLU and HardSwish wield better performance. We observe that ReLU can hardly provide any speedup over GeLU on iPhone 12 with CoreMLTools, while HardSwish is significantly slower than ReLU and GeLU. We draw a conclusion that the activation function can be selected on a case-by-case basis depending on the specific hardware and compiler. In this work, we use GeLU to provide better performance than ReLU while executing faster. For a fair comparison, we modify inefficient operators in other works according to the supports from iPhone 12 and CoreMLTools, *e.g.*, report LeViT latency by changing HardSwish to GeLU.

**Dimension Consistent Design.** We perform the latency analysis on the Nvidia A100 GPU to show that the proposed dimension-consistent (D-C) design is beneficial besides the iPhone. We deploy the PyTorch models with batch size 64 as ONNX format and use TensorRT to compile and benchmark the latency. We report the latency results averaged over 1,000 runs in Tab. 4. For the non-D-C design, we revert the proposed Meta3D block into 4D implementation, where linear projections and MLPs are all implemented with CONV1×1-BN instead of 3D-Linear layers, and reshaping operations become necessary in order to perform multi-head self-attention. With this configuration, attention blocks can be arbitrarily placed along with Meta4D blocks without following dimension-consistent design, while frequent reshaping is introduced. We conduct the comparison on the following two models, both with a D-C version and a non-D-C one with the exact same computation complexity:

- EfficientFormer-L7, which has 8 attention blocks.

- DummyNet, a handcrafted dummy model with a total of 16 attention blocks.

As can be seen from Tab. 4, the proposed dimension-consistent design achieves faster inference speed than the non-dimension-consistent design for both EfficientFormer-L7 and the DummyNet.

**Latency Driven Slimming.** Besides the hardware-efficient architecture design, it is still crucial to find appropriate depth and width configurations for the model to achieve satisfactory performance. To understand the benefits of our latency driven slimming, we randomly sample networks from our search space that have the same computation, *i.e.*, 1.3 GMACs, as our searched model EfficientFormer-L1. The sampled networks are denoted as Random 1 to Random 5, which are either deeper and narrower, or shallower and wider than EfficientFormer-L1. We train the sampled models on ImageNet-1K with the same training recipe as EfficientFormer-L1. The comparison between these models is shown in Tab. 5. As can be seen, our searched EfficientFormer-L1 has better latency or higher top-1 accuracy on ImageNet-1K than the randomly sampled networks, proving the advantages of our proposed latency driven slimming.

## C   Analysis of Hardware Utilization

EfficientFormer improves the latency vs. accuracy trade-off through better architecture design so that higher hardware utilization is achieved. To understand hardware utilization, we employ throughput in TFLOPS (Tera FLOPs per Second) as the evaluation metric, which is calculated by model computation cost (FLOPs) divided by execution time. Models with higher throughput (TFLOPS) better exploit the computation power of the hardware.

Table 3: Ablation analysis for the design choice on EfficientFormer-L3. V1-5 refers to variants with different operator selections.

| Model | CONV stem | Norm. | Activation | MHSA Search | Top-1 | Latency (ms) |
|---|---|---|---|---|---|---|
| EfficientFormer | ✓ | BN | GeLU | ✓ | 82.4 | 3.0 |
| V1 | ✓ | BN | GeLU | | 80.3 | 2.5 |
| V2-GN | ✓ | GN | GeLU | | 80.6 | 3.0 |
| V2-LN | ✓ | LN | GeLU | | 80.3 | 3.7 |
| V3 | ✓ | BN | ReLU | | 79.3 | 2.5 |
| V4 | ✓ | BN | HardSwish | | 80.3 | 32.4 |
| V5 | | BN | GeLU | | 79.6 | 5.8 |

Table 4: Analysis of dimension-consistent (D-C) design vs. non-D-C placement of attention blocks. The latency (ms) is measured on the Nvidia A100 GPU with TensorRT.

| Model | D-C | TensorRT-A100 (ms) |
|---|---|---|
| EfficientFormer-L7 | Y | 30.67 |
| EfficientFormer-L7 | N | 34.73 |
| DummyNet | Y | 7.07 |
| DummyNet | N | 11.93 |

Table 5: Analysis of latency driven slimming. All random networks are trained with the same training strategy as the EfficientFormer-L1 on ImageNet-1K. The latency is obtained on iPhone 12 NPU with CoreMLTools.

| Model | GMACs | Latency (ms) | Top-1 (%) |
|---|---|---|---|
| EfficientFormer-L1 | 1.3 | 1.6 | 79.2 |
| Random 1 | 1.3 | 1.6 | 77.8 |
| Random 2 | 1.3 | 1.7 | 78.3 |
| Random 3 | 1.3 | 1.5 | 74.7 |
| Random 4 | 1.3 | 1.6 | 73.3 |
| Random 5 | 1.3 | 1.5 | 76.7 |

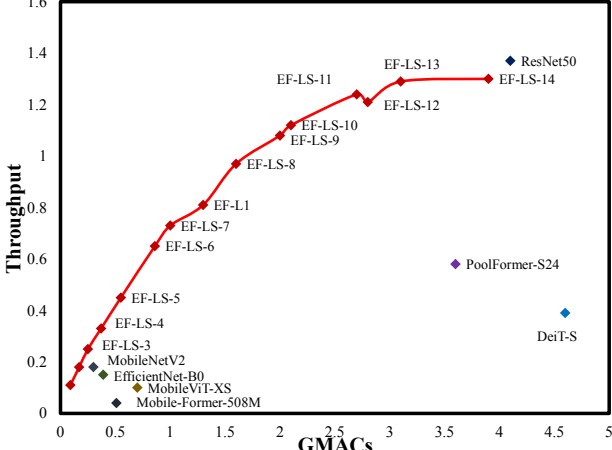

Figure 4: Analysis of hardware utilization on iPhone 12 NPU.

Table 6: Architecture details of EfficientFormer. $Exp$ refers to the expansion ratio of the MLP block. $D_{QK}$ is the dimension of Queries and Keys.

| Stage | Resolution | Type | Config | EfficientFormer L1 | EfficientFormer L3 | EfficientFormer L7 |
|---|---|---|---|---|---|---|
| stem | $\frac{H}{2} \times \frac{W}{2}$ | Patch Embed. | Patch Size | $k = 3 \times 3, s = 2$ | | |
| | | | Embed. Dim. | 24 | 32 | 48 |
| | $\frac{H}{4} \times \frac{W}{4}$ | Patch Embed. | Patch Size, | $k = 3 \times 3, s = 2$ | | |
| | | | Embed. Dim. | 48 | 64 | 96 |
| 1 | $\frac{H}{4} \times \frac{W}{4}$ | $MB^{4D}$ | Token Mixer | Pool | | |
| | | | $\begin{bmatrix} Embed. & Kernel \\ Stride & Exp \end{bmatrix}$ | $\begin{bmatrix} 48 & 3 \\ 1 & 4 \end{bmatrix} \times 3$ | $\begin{bmatrix} 64 & 3 \\ 1 & 4 \end{bmatrix} \times 4$ | $\begin{bmatrix} 96 & 3 \\ 1 & 4 \end{bmatrix} \times 6$ |
| 2 | $\frac{H}{8} \times \frac{W}{8}$ | Patch Embed. | Patch Size | $k = 3 \times 3, s = 2$ | | |
| | | | Embed. Dim. | 96 | 128 | 192 |
| | | $MB^{4D}$ | Token Mixer | Pool | | |
| | | | $\begin{bmatrix} Embed. & Kernel \\ Stride & Exp \end{bmatrix}$ | $\begin{bmatrix} 96 & 3 \\ 1 & 4 \end{bmatrix} \times 2$ | $\begin{bmatrix} 128 & 3 \\ 1 & 4 \end{bmatrix} \times 4$ | $\begin{bmatrix} 192 & 3 \\ 1 & 4 \end{bmatrix} \times 6$ |
| 3 | $\frac{H}{16} \times \frac{W}{16}$ | Patch Embed. | Patch Size | $k = 3 \times 3, s = 2$ | | |
| | | | Embed. Dim. | 224 | 320 | 384 |
| | | $MB^{4D}$ | Token Mixer | Pool | | |
| | | | $\begin{bmatrix} Embed. & Kernel \\ Stride & Exp \end{bmatrix}$ | $\begin{bmatrix} 224 & 3 \\ 1 & 4 \end{bmatrix} \times 6$ | $\begin{bmatrix} 320 & 3 \\ 1 & 4 \end{bmatrix} \times 12$ | $\begin{bmatrix} 384 & 3 \\ 1 & 4 \end{bmatrix} \times 8$ |
| 4 | $\frac{H}{32} \times \frac{W}{32}$ | Patch Embed. | Patch Size | $k = 3 \times 3, s = 2$ | | |
| | | | Embed. Dim. | 448 | 512 | 768 |
| | | $MB^{4D}$ | Token Mixer | Pool | | |
| | | | $\begin{bmatrix} Embed. & Kernel \\ Stride & Exp \end{bmatrix}$ | $\begin{bmatrix} 448 & 3 \\ 1 & 4 \end{bmatrix} \times 3$ | $\begin{bmatrix} 512 & 3 \\ 1 & 4 \end{bmatrix} \times 3$ | $\begin{bmatrix} 768 & 3 \\ 1 & 4 \end{bmatrix} \times 0$ |
| | $\frac{HW}{32^2}$ | $MB^{3D}$ | Token Mixer | MHSA | | |
| | | | $\begin{bmatrix} Embed. & D_{QK} \\ Heads & Exp \end{bmatrix}$ | $\begin{bmatrix} 448 & 32 \\ 8 & 4 \end{bmatrix} \times 1$ | $\begin{bmatrix} 512 & 32 \\ 8 & 4 \end{bmatrix} \times 3$ | $\begin{bmatrix} 768 & 32 \\ 8 & 4 \end{bmatrix} \times 8$ |

To fairly compare with baseline models under different computation complexity, we **L**inearly **S**cale the depth and width of EfficientFormer-L1 to obtain a series of models (EfficientFormer-LS-1 to EfficientFormer-LS-14), with the number of parameters ranging from 1.1M to 31.3M and MACs from 0.09G to 3.9G, and benchmark the latency and utilization on iPhone 12 NPU.

As in Fig. 4, super-tiny models still run at about 1ms, such as EfficientFormer-LS-{1, 2, 3}, where the throughput is low and the hardware is not fully exploited. Data processing and transferring become the bottleneck. As a result, making the model super small with sacrificed accuracy is less valuable. In contrast, our 1.3GMACs model, EfficientFormer-L1 lies at a sweet point, enjoying fast inference speed (1.6ms) while maintaining high accuracy.

Furthermore, we can observe that EfficientFormer variants outperform both CNNs and ViTs in hardware utilization across different computation complexity levels. For instance, at 4-GMACs level, EfficientFormer-LS-14 outperforms DeiT-S by 3.3× higher TFLOPS and outperforms PoolFormer by 2.2×, achieving comparable throughput to ResNet50. In the lightweight domain, EfficientFormer-LS-4 has 2.2× higher TFLOPS than EfficientNet-B0. We demonstrate that with the proposed hardware-friendly design, EfficientFormer naturally has better hardware utilization.

# D   Architecture of EfficientFormers

The detailed network archiecture for EfficientFormer-L1, EfficientFormer-L3, and EfficientFormer-L7 is provided in Tab. 6. We report the resolution and number of blocks for each stage. In addition, the width of EfficientFormer is specified as the embedding dimension (Embed. Dim.). As for the MHSA block, the dimension of Query and Key is provided, and we employ eight heads for all EfficientFormer variants. MLP expansion ratio is set as default (4), as in most ViT arts [3].