# OpenReview forum: "EfficientFormer: Vision Transformers at MobileNet Speed"
_NeurIPS.cc/2022/Conference — NeurIPS 2022 Accept_

### Official Review · Reviewer_mbmg · 2022-07-12

**Rating:** 6
**Confidence:** 3
**Soundness:** 3 good
**Presentation:** 3 good
**Contribution:** 3 good

**Summary:**

The paper benchmarks the latency of recent efficient vision transformer designs on iPhone 12 through CoreML support and makes observations on the latency vs. accuracy trade-off of different operators. The authors then proposed a super-net search space that bakes in multiple latency-favorable designs and performs gradient-based architecture search followed by a heuristic latency-driven slimming process to obtain sub-net under given latency constraint. The authors evaluate the searched architectures as the backbone across image classification, object detection, and semantic segmentation tasks.

**Questions:**

Looking at the comparison in Table 1, at a given latency, EfficientViT tends to have a higher parameter count or GMACs (i.e. EfficientViT-L1 vs. MobileNetV2, EfficientViT-L7 vs. MobileViT-XS). From reading the paper, it seems the improvement could come from two sources: 1) better hardware utilization so that a more complex model can run under the same latency constraint (latency analysis guided design space), and 2) when hardware utilization saturates, finding architecture with better compute distribution over operators for higher accuracy (NAS + latency-driven slimming). Is there a way to perform ablation and quantify the performance gain from each source? Understanding the contribution from 1) would be valuable for practitioners optimizing architecture for specific hardware, while understanding 2) will shed light on the value of the proposed design space and search algorithm.


**Limitations:**

The authors clearly addressed some potential limitations of the work: 1) Some observations and subsequent design decisions might be hardware and software dependent; 2) The NAS procedure, specifically the latency-driven slimming procedure is less involved and could be a direction for future exploration.

**Strengths And Weaknesses:**

- *Originality:* To the best of my knowledge, the paper adequately cites related works such as MetaFormer and outlines the differences. While the authors mention that the design space is partly inspired by MetaFormer, the latency vs. accuracy trade-off improvement is novel and largely driven by new insights through latency benchmark on actual hardware.
- *Quality*: The submission is technically solid in building the EfficientViT design space based on observations from latency analysis on real hardware. The authors clearly addressed two relevant limitations that come to mind when reading the paper: 1) limited insight on how the observations and designs hold across other hardware, and 2) the latency-driven slimming procedure could invite further investigation. Testing the searched architectures across different tasks is a good practice as well.
- *Clarity*: The submission is clearly written and well organized. The observations from latency analysis naturally build up into respective considerations for the design space.
- *Significance:* The paper provides comprehensive empirical insights on optimizing vision transformer design space for a given hardware, and achieving MobileNetV2 level latency is meaningful for practitioners. While the latency analysis workflow is general and could serve as a good example of devising design decisions for other hardware, it is not immediately clear how well the insights and the design space will transfer.

---

> ### Author Response · Authors · 2022-08-02
> **Author Responses to Reviewer mbmg (Part 4/4)**
>
> **Q2. Ablation on hardware utilization and the searching algorithm.**
>
> **2.2. Analysis of latency driven slimming.**
>
> Besides the hardware-efficient architecture design, it is still crucial to find appropriate depth and width configurations for the model to achieve satisfactory performance. To understand the benefits of our latency driven slimming, we randomly sample networks from our search space that have the same computation, i.e.,1.3 GMACs, as our searched model EfficientViT-L1. The sampled networks are denoted as Random 1 to Random 5, which are either deeper and narrower, or shallower and wider than EfficientViT-L1. We train the sampled models on ImageNet-1K with the same training recipe as EfficientViT-L1. The comparison between these models is shown in the following table (Table H). As can be seen, our searched EfficientViT-L1 has better latency or higher top-1 accuracy on ImageNet-1K than the randomly sampled networks, proving the advantages of our proposed latency driven slimming.
>
> >**Table H. Analysis of latency driven slimming. All random networks are trained with the same training strategy as the EfficientViT-L1 on ImageNet-1K. The latency is obtained using iPhone 12 with CoreMLTools.**
> | Model | GMACs | Latency (ms) | Top-1 (%) |
> |:---:|:---:|:---:|:---:|
> | **EfficientViT-L1** | **1.3** | **1.6** | **79.2** |
> | Random 1 | 1.3 | 1.6 | 77.8 |
> | Random 2 | 1.3 | 1.7 | 78.3 |
> | Random 3 | 1.3 | 1.5 | 74.7 |
> | Random 4 | 1.3 | 1.6 | 73.3  |
> | Random 5 | 1.3 | 1.5 | 76.7 |
>
> ---
>
> **References:**
>
> [a] https://www.nvidia.com/en-us/data-center/a100
>
> [b] https://onnx.ai/
>
> [c] https://developer.nvidia.com/tensorrt
>
> [d] https://catalog.ngc.nvidia.com/orgs/nvidia/containers/tensorrt
>
> [e] https://developer.android.com/ndk/guides/neuralnetworks

---

> ### Author Response · Authors · 2022-08-02
> **Author Responses to Reviewer mbmg (Part 3/4)**
>
> **Q2. Ablation on hardware utilization and the searching algorithm.**
>
> First of all, thanks for the insightful comments! We totally agree with you that our model improves the latency vs. accuracy trade-off through (i) better architecture design so that higher hardware utilization is achieved, and (ii) leveraging latency driven slimming to find fast models while maintaining accuracy. As suggested, we provide a detailed analysis as follows to quantify the performance gain from each of them.
>
> ---
>
> **2.1. Analysis of hardware utilization.**
>
> To understand the hardware utilization, we employ throughput in TFLOPS (Tera FLOPs per Second) as the evaluation metric, which is calculated by model computation cost (FLOPs) divided by execution time. Models with higher throughput (TFLOPS) better exploit the computation power of the hardware.
>
> To fairly compare with baseline models under different computation complexity, we linearly scale the depth and width of EfficientViT-L1 to obtain a series of models (EfficientViT-LS-1 to EfficientViT-LS-14), with the number of parameters ranging from 1.1M to 31.3M and MACs from 0.09G to 3.9G, and benchmark the latency and utilization on iPhone 12.
>
> As in the following table (Table G, rows are sorted based on GMACs in descending order), super-tiny models still run at about 1ms, such as EfficientViT-LS-1, EfficientViT-LS-2, and EfficientViT-LS-3, where the throughput is low and the hardware is not fully exploited. Data processing and transferring become the bottleneck. As a result, making the model super small with sacrificed accuracy is less valuable. In contrast, our 1.3GMACs model, EfficientViT-L1 lies at a sweet point, enjoying fast inference speed (1.6ms) while maintaining high accuracy.
>
> Furthermore, we can observe that EfficientViT variants outperform both CNNs and ViTs in hardware utilization across different computation complexity levels.
> For instance, at 4-GMACs level, EfficientViT-LS-14 outperforms DeiT-S by 3.3$\times$ higher TFLOPS and outperforms PoolFormer by 2.2$\times$, achieving comparable throughput to ResNet50.
> In the lightweight domain, EfficientViT-LS-4 has 2.2$\times$ higher TFLOPS than EfficientNet-B0.
> We demonstrate that with the proposed hardware-friendly design, EfficientViT naturally has better hardware utilization.
>
> Due to the format constraints for author response, the throughput data shall be visualized in graph view in the revision.
>
> >**Table G. Analysis of hardware utilization on iPhone 12.**
> | Model | Params (M) | GMACs | Latency (ms) | Throughput (TFLOPS) |
> |:---:|:---:|:---:|:---:|:---:|
> | DeiT-S | 22.5 | 4.6 | 11.8 | 0.39 |
> | ResNet50 | 25.5 | 4.1 | 3.0 | 1.37 |
> | EfficientViT-LS-14 | 31.3 | 3.9 | 3.0 | 1.30 |
> | PoolFormer-S24 | 21.0 | 3.6 | 6.2 | 0.58 |
> | EfficientViT-LS-13 | 23.5 | 3.1 | 2.41 | 1.29 |
> | EfficientViT-LS-12 | 20.8 | 2.8 | 2.31 | 1.21 |
> | EfficientViT-LS-11 | 19.7 | 2.7 | 2.17 | 1.24 |
> | EfficientViT-LS-10 | 15.9 | 2.1 | 1.88 | 1.12 |
> | EfficientViT-LS-9 | 15.8 | 2.0 | 1.85 | 1.08 |
> | EfficientViT-LS-8 | 12.1 | 1.6 | 1.65 | 0.97 |
> | **EfficientViT-L1** | **12.3** | **1.3** | **1.60** | **0.81** |
> | EfficientViT-LS-7 | 8.2 | 1.0 | 1.37 | 0.73 |
> | EfficientViT-LS-6 | 6.9 | 0.86 | 1.33 | 0.65 |
> | MobileViT-XS | 2.3 | 0.70 | 7.20 | 0.10 |
> | EfficientViT-LS-5 | 5.0 | 0.55 | 1.23 | 0.45 |
> | MobileFormer | 14.0 | 0.51 | 13.22 | 0.04 |
> | EfficientNet-B0 | 5.3 | 0.39 | 2.71 | 0.15 |
> | EfficientViT-LS-4 | 3.8 | 0.37 | 1.13 | 0.33 |
> | MobileNetV2 | 3.5 | 0.30 | 1.70 | 0.18 |
> | EfficientViT-LS-3 | 2.8 | 0.25 | 1.02 | 0.25 |
> | EfficientViT-LS-2 | 2.0 | 0.17 | 0.95 | 0.18 |
> | EfficientViT-LS-1 | 1.1 | 0.09 | 0.85 | 0.11 |

---

> ### Author Response · Authors · 2022-08-02
> **Author Responses to Reviewer mbmg (Part 2/4)**
>
> **Q1. Transfer the insights and design space to other hardware and compilers.**
>
> We thank the reviewer for the suggestion. Here we provide additional results by deploying our and other models on different hardware and compilers. We show the average latency of over 1,000 runs for the following hardware and compilers.
> - Nvidia A100 GPU with TensorRT. We run the latency analysis on the Nvidia A100 GPU [a] with batch size 64. The Pytorch models are saved into the ONNX format [b] and compiled with TensorRT [c]. We report two latency results from TensorRT in the following table (Table E). One is the computing time on GPU (TRT-A100-GPU), and the other is the total walltime that includes the time for data transfer (TRT-A100-Total). We use the latest Nvidia software environment for the experiments [d].
> - iPhone CPU with CoreMLTools. We benchmark the latency for models by only using the CPU in the iPhone 12. The models are deployed by CoreMLTools.
> - Google Pixel 6 with NNAPI. We report the model latency on android devices. We utilize the Google Pixel 6 with NNAPI [e] for model compiling. Please note that NNAPI does not well support the GeLU, so we replace the GeLU with HardSwish in all models that include GeLU for a fair comparison. Models are converted into TensorFlow Lite format and deployed using NNAPI. Due to the compatibility issue of NNAPI, many converted models can not successfully run on Pixel 6. Therefore, we only report the latency for the models that can. We leave the support for more baseline models on Google Pixel as future work.
>
> The following tables (Table E and Table F) report the latency analysis on the Nvidia A100 GPU with TensorRT, iPhone CPU with CoreMLTools, and Pixel 6 with NNAPI for the models trained on ImageNet-1K for the classification task. We can see our model still achieves decent latency vs. accuracy trade-off improvement on different hardware and compilers.
>
> For example, compared with the CNN models, EfficientViT-L1 runs faster (38% faster on Nvidia A100 GPU Computing and 21% faster on iPhone CPU) than EfficientNet-B0 while achieving 2.1% higher top-1 accuracy. For the models with high performance (>83% top-1), EfficientFormer-L7 runs much faster (4.6$\times$ faster on Nvidia A100 GPU Computing and 3.8$\times$ faster on iPhone CPU) than EfficientNet-B5.
>
> Compared to ViTs and their variants, EfficientViT-L1 has 4.4% higher top-1 accuracy than MobileViT-XS and runs much faster across different hardware and compilers (1.9$\times$ faster on Nvidia A100 GPU Computing, 2.3$\times$ faster on iPhone CPU, and 10.4$\times$ faster on Pixel 6), and has 4.7% higher accuracy than DeiT-T while being 8.3$\times$ faster on Pixel 6. Also, EfficientViT-L3 achieves 1% higher top-1 accuracy than PoolFormer-S36, while being 3$\times$ faster on Nvidia A100 GPU and 2.8$\times$ faster on iPhone CPU. The results on different hardware and compilers demonstrate the advantageous performance of our models.
>
> >**Table E. Comparison results on ImgeNet-1K. The latency (ms) is measured on the Nvidia A100 GPU with TensorRT (TRT-A100-GPU and TRT-A100-Total) and iPhone 12 CPU with CoreMLTools. ‘/’ denotes the model is not well supported by the hardware and compiler.**
> | Model | Train epoch | Top-1 | TRT-A100-GPU(ms) | TRT-A100-Total (ms) | iPhone CPU (ms) |
> |:---:|:---:|:---:|:---:|:---:|:---:|
> | **EfficientViT-L1** | **300** | **79.2** | **6.17** | **9.33** | **11.5** |
> | **EfficientViT-L1** | **450** | **79.9** | **6.17** | **9.33** | **11.5** |
> | **EfficientViT-L3** | **300** | **82.4** | **13.94** | **17.10** | **28.2** |
> | **EfficientViT-L7** | **300** | **83.3** | **30.67** | **33.83** | **67.7** |
> | MobileNetV2 | 300 | 71.9 | 4.97 | 8.13 | 8.0 |
> | MobileNetV2 x 1.4 | 300 | 74.7 | 7.32 | 10.47 | 10.7 |
> | EfficientNet-B0 | 350 | 77.1 | 9.99 | 13.15 | 14.5 |
> | EfficientNet-B3 | 350 | 81.6 | 35.03 | 40.67 | 52.6 |
> | EfficientNet-B5 | 350 | 83.6 | 141.00 | 153.97 | 258.8 |
> | ResNet50 | 300 | 78.5 | 9.02 | 12.17 | 29.4 |
> | ResMLP-S24 | 300 | 79.4 | 17.35 | 20.51 | 40.2 |
> | DeiT-T | 300 | 74.5 | 7.08 | 10.24 | 16.7 |
> | DeiT-Small | 300 | 81.2 | 15.45 | 18.60 | 41.0 |
> | PVT-small | 300 | 79.8 | 23.75 | 26.91 | 89.5 |
> | T2T-ViT-14 | 310 | 81.5 | 20.99 | 24.15 | / |
> | Swin-Tiny | 300 | 81.3 | 21.99 | 25.15 | / |
> | PoolFormer-s12 | 300 | 77.2 | 14.52 | 19.44 | 59.0 |
> | PoolFormer-s24 | 300 | 80.3 | 28.22 | 33.10 | 126.7 |
> | PoolFormer-s36 | 300 | 81.4 | 41.21 | 46.03 | 192.6 |
> | Mobile-Former-508m | 450 | 79.3 | 14.58 | 17.74 | 22.2 |
> | MobileViT-XS | 300 | 74.8 | 11.65 | 14.81 | 26.5 |
>
> >**Table F. Comparison results on ImgeNet-1K. The latency (ms) is measured on Google Pixel 6 with NNAPI (Android - Pixel 6).**
> | Model | Train epoch | Top-1 | Android - Pixel 6 (ms) |
> |:---:|:---:|:---:|:---:|
> | **EfficientViT-L1** | **300** | **79.2** | **7.89** |
> | DeiT-T | 300 | 74.5 | 65.60 |
> | MobileViT-XS | 300 | 74.8 | 82.49 |

---

> ### Author Response · Authors · 2022-08-02
> **Author Responses to Reviewer mbmg (Part 1/4)**
>
> **We thank the reviewer for the positive feedback and valuable suggestions. We appreciate that the reviewer acknowledges our work introduces novel improvement for latency vs. accuracy trade-off; proposes technically solid EfficientViT design space and general latency analysis workflow; provides comprehensive insights on optimizing vision transformers; achieves meaning latency for practitioners; and the paper is clearly written and well organized. In the following, we provide results on transferring the design space to other hardware and compilers (Q1) and conduct an ablation study on the hardware utilization and searching algorithm (Q2).**

---

> ### Author Response · Authors · 2022-08-08
> **Author Responses to Reviewer mbmg**
>
> Dear Reviewer mbmg,
>
> Thanks again for your time and reviewing efforts to help improve our work! We appreciate your positive rating and insightful comments.
>
> As a kind reminder, we provide suggested results and comparisons in the authors' response, including the demonstration of the advantageous performance of EfficientViT on other hardware and compilers (Nvidia A100 with TensorRT, iPhone CPU with CoreML, and Android device with NNAPI), and ablations on hardware utilization and the latency-driven search algorithm. We hope our responses have addressed your concerns.
>
> Best,
>
> Authors

---

### Official Review · Reviewer_X2kc · 2022-07-14

**Rating:** 5
**Confidence:** 5
**Soundness:** 3 good
**Presentation:** 3 good
**Contribution:** 2 fair

**Summary:**

In this paper, the authors propose a new ViT-based architecture named EfficientViT. In detail, this paper tries to build hybrid designs for MobileNet block and ViT architectures. Finally, this algorithm performs a latency-driven slimming for a series of final models.

**Questions:**

As shown in the Strengths And Weaknesses.

**Limitations:**

I do not find any limitations and potential negative societal impact of this work.

**Strengths And Weaknesses:**

Strengths:
1. This paper is easy to read.
2. The proposed observations in this paper are interesting and useful for the designation of architectures.

Weaknesses:
1. Many papers focus on the combination of CNNs and Transformers.
2. The proposed MB blocks have similar architectures to some existing papers.
3. This paper does not compare with some baseline methods[1][2], and these methods perform better than this paper.

[1] Mobile-Former: Bridging MobileNet and Transformer
[2] CSWin Transformer: A General Vision Transformer Backbone with Cross-ShapedWindows

---

> ### Author Response · Authors · 2022-08-02
> **Author Responses to Reviewer X2kc**
>
> **We thank the reviewer for the valuable feedback. We appreciate that the reviewer acknowledges our paper is easy to follow, interesting, and useful for the designation of architectures. We address the concerns in the following. We hope our response can further demonstrate the strengths of our method.**
>
> ---
>
> **Q1. Many papers focus on the combination of CNNs and Transformers.**
>
> We thank the reviewer for the comment. If we understand correctly, the concern is that our work combines CNNs and Transformers, which is a strategy that is studied by some papers. In the following, we would like to kindly clarify our method and try to alleviate the concern.
>
> First, we would like to kindly mention that our work targets the efficient deployment of pure transformer models on mobile devices, instead of incorporating MobileNet blocks or depth-wise convolutions to reduce computation costs. Consequently, EfficientViT is built with token mixers (either global attention or local pooling) and MLP blocks, which is a standard transformer architecture. Based on our experimental results, without the integration of lightweight MobileNet blocks, EfficientViT still outperforms hybrid design in terms of speed-accuracy trade-off, which we humbly think is the strength of our work. Detailed discussion and comparisons can be found in Section 2 and 5.1.
>
> Second, we would like to kindly explain that our network architecture design is still based on the transformer architecture, and the usage of CNN-like parts in the 4D partition of EffiicientViT is to align the feature dimension for efficient inference. For example, our CONV stem reduces the patch size of classic ViTs, which is better supported by edge devices. Similarly, we implement linear projections and MLP blocks through CONV1x1 layers, such that the dimension is ensured to be consistent in the 4D partition. Different from existing works, we investigate how the data dimension (4D CONV or 3D Linear) in transformers affects hardware efficiency, and propose a dimension consistent design that enables ultra-fast inference on edge devices, as demonstrated in Figure 2 and Section 3, which is one of the contributions of this work.
>
> ---
>
> **Q2. The proposed MB blocks have similar architectures to some existing papers.**
>
> In fact, our work targets the acceleration of standard vision transformers for efficient mobile deployment. Therefore, we design the MetaBlocks using the commonly adopted token mixer and regular MLP design. We do not aim to propose new complicated operations, such as shifted-window attention in Swin, to boost model accuracy. Instead, we develop the most hardware-friendly design strategies for ViTs for fast inference, including hardware-friendly operators (Section 3), dimension-consistent design (Section 4.1), and latency-driven architecture search (Section 4.2), while maintaining high performance.
>
> We humbly think that following standard transformer architecture is not a weakness of our work. On the contrary, boosting the standard architecture to achieve significantly better speed and accuracy is an advantage and novelty of our work.
>
> ---
>
> **Q3. Comparisons with Mobile-Former and CSwin.**
>
> We thank the reviewer for suggesting the comparison with recent arts, Mobile-Former and CSwin. The comparison with the two works is shown in the following table (Table D), demonstrating better latency and performance of our models over Mobile-Former and CSwin.
>
> Mobile-Former released the official models recently (after the submission of this draft). We perform the speed comparison on iPhone 12 and Nvidia A100 GPU, and will add these results in the revision. Under the same training resources, i.e., 450 epochs, our EfficientViT-L1 achieves 0.6% higher top-1 accuracy on ImageNet-1K than Mobile-Former-508M, while being 8.3$\times$ faster on iPhone 12 and 2.4$\times$ faster on A100 GPU. CSwin adopts a complicated token mixer such as the cross-shaped window. Such operation is not supported by most compilers on mobile devices (as discussed on Line 122 in the main paper). Efficiently implementing the cross-shaped window in the mobile compiler is beyond the scope of this paper, thus we provide a comparison on the Nvidia A100 GPU. We linearly scale up EfficientViT-L3 to obtain EfficientViT-L3-LS that matches the computation cost of CSwin-T, achieving slightly higher accuracy and 1.4$\times$ faster speed.
> Thanks for the suggestions and we will properly discuss Mobile-Former and CSwin in the revised paper.
>
>
>
> >**Table D. Comparison results on ImgeNet-1K with Mobile-Former-508M and CSwin, deployed on iPhone 12 with CoreMLTools and Nvidia A100 GPU with TensorRT.**
> | Model | Train epoch | Top-1 | iPhone 12 (ms) | TRT-A100-GPU (ms) |
> |:---:|:---:|:---:|:---:|:---:|
> | **EfficientViT-L1** | **450** | **79.9** | **1.6** | **6.17** |
> | Mobile-Former-508M | 450 | 79.3 | 13.2 | 14.58 |
> | **EfficientViT-L3-LS** | **300** | **82.8** | **3.4** | **20.55** |
> | CSwin-T | 300 | 82.7 | Not supported | 28.70 |

---

> > ### Comment · Reviewer_X2kc · 2022-08-07
> > **Good.**
> >
> > Thanks for your rebuttal.
> >
> > I like your rebuttal with your experiments and explanations. I will change the final rating to 5.

---

> > > ### Author Response · Authors · 2022-08-08
> > > **To Reviewer X2kc**
> > >
> > > Dear Reviewer X2kc,
> > >
> > > Thank you so much for checking our responses and raising the score. It is our great pleasure to know our efforts have helped address your concerns!
> > >
> > > We appreciate your time and reviewing efforts to help improve our work. If you still have questions or concerns, we would sincerely like to know and will make the best of our efforts to resolve them within the open discussion period.
> > >
> > > Best,
> > >
> > > Authors

---

> ### Author Response · Authors · 2022-08-05
> **Author Responses to Reviewer X2kc**
>
> Dear Reviewer X2kc,
>
> We appreciate your time and reviewing efforts to help improve our work. Thanks!
>
> We follow your initial suggestions to provide additional results, such as the comparison with Mobile-Former and CSwin, to clarify the advantage of our model, especially on mobile devices.
> We also provide the differences between our work and others. We hope our response can help further demonstrate that our approach is crucial for designing mobile-friendly transformer architectures.
>
> As the deadline for the author-reviewer discussion is approaching, we would sincerely appreciate it if you could kindly let us know whether our response addressed your concerns, and please let us know if you have further questions. It will be our great pleasure if you would consider updating your review or score.
>
> Best,
>
> Authors

---

### Official Review · Reviewer_RfgL · 2022-07-19

**Rating:** 6
**Confidence:** 4
**Soundness:** 3 good
**Presentation:** 4 excellent
**Contribution:** 3 good

**Summary:**

This paper proposes a combination of several implementation details for vision transformers that can be evaluated efficiently (with low latency) on iphones. These details include:
- Clever management of the shape of the tensors to avoid expensive reshape operations.
- Using BN instead of LN to save latency by folding in BN parameters into the final model.
- Using GELU nonlinearity (eg. instead of swish)
and most importantly a pruning methodology to prune the least essential blocks by training a categorical gating mechanism via gradient descent.
The resulting network vastly outperforms previous solutions that have similar latency on both ImageNet classification and semantic segmentation.

**Questions:**

- Are the dimension-consistent layers relevant in other libraries/hardware, like pytorch, TF or jax on nvidia GPUs? (As the model was trained on  pytorch/GPU), this should not be hard to answer.

**Limitations:**

Obviously, the paper focuses on a very specific proprietary hardware/software stack, and therefore the applicability of the methods is somewhat limited. The paper does not reference very similar pruning methods presented in 2018 MorphNet paper.

**Strengths And Weaknesses:**

Originality: Weak-Medium.
The paper presents a combination of methods, most of which is well-studied and understood in different contexts:
- Shape management and alignment has been one of the most optimized aspect of highly tuned numerical linear algebra software for many decades. For example, this has been the focus of optimizing BLAS libraries over many decades.
- It is well-known that BN has a slight edge over other normalization methods when it comes to latency of the final model, due to the possibility of folding in the normalization parameters.
- It is surprising that swish is so inefficient on CoreML, but it seems like a special artifact of the current state of the library.
- While the pruning methodology seems very efficient, it resembles very much to MorphNet [Gordon, Ariel, et al. "Morphnet: Fast & simple resource-constrained structure learning of deep networks." CVPR. 2018], while failing to cite that prior work.

Quality: Medium
While it is impressive that 79% top-1 accuracy on ImageNet is possible within 1.2ms on an iphone, the papers sole focus on this particular library and hardware raises the question of the general applicability of the methods. Also several of the methods just circumvent the special deficiencies of the library (esp swish vs GELU accounts for a large parts of improvements over LeViT 256, which might become competitive with this single change.

Clarity: High
The paper is well-written with clear ablation analyses and explaining the motivation of all the applied techniques. The methods are evaluated on reasonable benchmarks on both classification and segmentation tasks.

Significance: Medium
This work gives a strong SoTA baseline for vision models on current IPhone hardware. However, it is unclear how/whether these methods generalize to other types of hardware/libraries. It seems unlikely that similar gains could be achieved on android systems, given the difference and maturity of the employed machine learning infrastructure.

---

> ### Author Response · Authors · 2022-08-02
> **Author Responses to Reviewer RfgL (Part 2/2)**
>
> **Q2.  Applicability of the models on more hardware and compilers.**
>
> In the main paper, we conduct the latency analysis on iPhone 12 NPU with all available computing resources and deploy models with CoreMLTools. Here we show more results on different hardware and compilers. The reported latency is averaged at over 1,000 runs.
> - Nvidia A100 GPU with TensorRT. We run the latency analysis on the Nvidia A100 GPU [a] with batch size 64. The Pytorch models are saved into the ONNX format [b] and compiled with TensorRT [c]. We report two latency results from TensorRT in the following table (Table B). One is the computing time on GPU (TRT-A100-GPU), and the other is the total walltime that includes the time for data transfer (TRT-A100-Total). We use the latest Nvidia software environment for the experiments [d].
> - iPhone CPU. We benchmark the latency for models by only using the CPU in the iPhone 12. The models are deployed by CoreMLTools.
> - Google Pixel 6 with NNAPI. As suggested by the reviewer, we also report the model latency on android devices. We utilize the Google Pixel 6 with NNAPI [e] for model compiling. Please note that NNAPI does not well support the GeLU, so we replace the GeLU with HardSwish in all models that include GeLU for a fair comparison. Models are converted into TensorFlow Lite format and deployed using NNAPI. Due to the compatibility issue of NNAPI, many converted models can not successfully run on Pixel 6. Therefore, we only report the latency for the models that can. We leave the support for more baseline models on Google Pixel as future work.
>
> The following tables (Table B and Table C) report the latency analysis on the Nvidia A100 GPU with TensorRT, iPhone CPU with CoreMLTools, and Pixel 6 with NNAPI for the models trained on the ImageNet-1K classification task. We can see our model still achieves decent latency vs. accuracy trade-off improvement on different hardware and compilers.
>
> For example, compared with the CNN models, EfficientViT-L1 runs faster (38% faster on Nvidia A100 GPU Computing and 21% faster on iPhone CPU) than EfficientNet-B0 while achieving 2.1% higher top-1 accuracy. For the models with high performance (>83% top-1), EfficientFormer-L7 runs much faster (4.6$\times$ faster on Nvidia A100 GPU Computing and 3.8$\times$ faster on iPhone CPU) than EfficientNet-B5.
>
> Compared to ViTs and their variants, EfficientViT-L1 has 4.4% higher top-1 accuracy than MobileViT-XS and runs much faster across different hardware and compilers (1.9$\times$ faster on Nvidia A100 GPU Computing, 2.3$\times$ faster on iPhone CPU, and 10.4$\times$ faster on Pixel 6), and has 4.7% higher accuracy than DeiT-T while being 8.3$\times$ faster on Pixel 6. Also, EfficientViT-L3 achieves 1% higher top-1 accuracy than PoolFormer-S36, while being 3$\times$ faster on Nvidia A100 GPU and 2.8$\times$ faster on iPhone CPU. The results on different hardware and compilers demonstrate the advantageous performance of our models.
>
> >**Table B. Comparison results on ImgeNet-1K. The latency (ms) is measured on the Nvidia A100 GPU with TensorRT (TRT-A100-GPU and TRT-A100-Total) and iPhone 12 CPU with CoreMLTools. ‘/’ denotes the model is not well supported by the hardware and compiler.**
> | Model | Train epoch | Top-1 | TRT-A100-GPU(ms) | TRT-A100-Total (ms) | iPhone CPU (ms) |
> |:---:|:---:|:---:|:---:|:---:|:---:|
> | **EfficientViT-L1** | **300** | **79.2** | **6.17** | **9.33** | **11.5** |
> | **EfficientViT-L1** | **450** | **79.9** | **6.17** | **9.33** | **11.5** |
> | **EfficientViT-L3** | **300** | **82.4** | **13.94** | **17.10** | **28.2** |
> | **EfficientViT-L7** | **300** | **83.3** | **30.67** | **33.83** | **67.7** |
> | MobileNetV2 | 300 | 71.9 | 4.97 | 8.13 | 8.0 |
> | MobileNetV2 x 1.4 | 300 | 74.7 | 7.32 | 10.47 | 10.7 |
> | EfficientNet-B0 | 350 | 77.1 | 9.99 | 13.15 | 14.5 |
> | EfficientNet-B3 | 350 | 81.6 | 35.03 | 40.67 | 52.6 |
> | EfficientNet-B5 | 350 | 83.6 | 141.00 | 153.97 | 258.8 |
> | ResNet50 | 300 | 78.5 | 9.02 | 12.17 | 29.4 |
> | ResMLP-S24 | 300 | 79.4 | 17.35 | 20.51 | 40.2 |
> | DeiT-T | 300 | 74.5 | 7.08 | 10.24 | 16.7 |
> | DeiT-Small | 300 | 81.2 | 15.45 | 18.60 | 41.0 |
> | PVT-small | 300 | 79.8 | 23.75 | 26.91 | 89.5 |
> | T2T-ViT-14 | 310 | 81.5 | 20.99 | 24.15 | / |
> | Swin-Tiny | 300 | 81.3 | 21.99 | 25.15 | / |
> | PoolFormer-s12 | 300 | 77.2 | 14.52 | 19.44 | 59.0 |
> | PoolFormer-s24 | 300 | 80.3 | 28.22 | 33.10 | 126.7 |
> | PoolFormer-s36 | 300 | 81.4 | 41.21 | 46.03 | 192.6 |
> | Mobile-Former-508m | 450 | 79.3 | 14.58 | 17.74 | 22.2 |
> | MobileViT-XS | 300 | 74.8 | 11.65 | 14.81 | 26.5 |
>
> >**Table C. Comparison results on ImgeNet-1K. The latency (ms) is measured on Google Pixel 6 with NNAPI (Android - Pixel 6).**
> | Model | Train epoch | Top-1 | Android - Pixel 6 (ms) |
> |:---:|:---:|:---:|:---:|
> | **EfficientViT-L1** | **300** | **79.2** | **7.89** |
> | DeiT-T | 300 | 74.5 | 65.60 |
> | MobileViT-XS | 300 | 74.8 | 82.49 |
>
> References:
>
> [e] https://developer.android.com/ndk/guides/neuralnetworks

---

> ### Author Response · Authors · 2022-08-02
> **Author Responses to Reviewer RfgL (Part 1/2)**
>
> **We thank the reviewer for the positive feedback and thoughtful comments. We appreciate the reviewer's acknowledgment that the paper proposes efficient methods, achieves impressive results on ImageNet with low latency, gives a strong SOTA baseline for vision models on iPhone, and is well-written with clear ablation analysis. We thank the reviewer for mentioning MorphNet [Gordon, Ariel, et al., 2018], which is a relevant and insightful work. We will cite and discuss it in the revised paper. In the following, we validate dimension-consistent design on other platforms (Q1) and provide more results on different hardware and compilers (Q2).**
>
> ---
>
> **Q1. Validate dimension-consistent design on other libraries/hardware (Nvidia GPU).**
>
> Thanks for the suggestion. We perform the latency analysis on the Nvidia A100 GPU [a] to show that the proposed dimension-consistent (D-C) design is beneficial besides the iPhone. We save the Pytorch models with batch size as 64 into the onnx format [b] and use TensorRT [c] to compile and benchmark the latency. We report two latency results from TensorRT averaged over 1,000 runs in the following table (Table A). One is the computing time on GPU (TRT-A100-GPU), and the other is the total walltime that includes the time for data transfer (TRT-A100-Total). We use the latest software version released by Nvidia for the experiments [d].
>
> For the non-D-C design, we revert the proposed Meta3D block into 4D implementation, where linear projections and MLPs are all implemented with CONV1x1-BN instead of 3D-Linear layers, and reshaping operations become necessary in order to perform multi-head self-attention. With this configuration, attention blocks can be arbitrarily placed along with Meta4D blocks without following dimension-consistent design, while frequent reshaping is introduced.
> We conduct the comparison on the following two models, both with a D-C version and a non-D-C one with the exact same computation complexity.
> - EfficientViT-L7, which has 8 attention blocks.
> - DummyNet, a handcrafted dummy model with a total of 16 attention blocks.
>
> As can be seen from Table A, our dimension-consistent design archives a faster inference speed than the non- dimension-consistent design for both EfficientViT-L7 and the DummyNet.
>
> >**Table A. Analysis of dimension-consistent (D-C) design vs. non-D-C placement of attention blocks. The latency (ms) is measured on the Nvidia A100 GPU with TensorRT (TRT-A100-GPU and TRT-A100-Total).**
> | Model | D-C | TRT-A100-GPU (ms) | TRT-A100-Total (ms) |
> |:---:|:---:|:---:|:---:|
> | EfficientViT-L7 | Y | 30.67 | 33.83 |
> | EfficientViT-L7 | N | 34.73 | 37.89 |
> | DummyNet | Y | 7.07 | 10.22 |
> | DummyNet | N | 11.93 | 15.09 |
>
> References:
>
> [a] https://www.nvidia.com/en-us/data-center/a100
>
> [b] https://onnx.ai/
>
> [c] https://developer.nvidia.com/tensorrt
>
> [d] https://catalog.ngc.nvidia.com/orgs/nvidia/containers/tensorrt

---

> ### Author Response · Authors · 2022-08-08
> **Author Responses to Reviewer RfgL**
>
> Dear Reviewer RfgL,
>
> Thanks again for your time and reviewing efforts to help improve our work! We appreciate your positive rating and insightful comments.
>
> As a kind reminder, we provide suggested results and comparisons in the authors' response, including the generalization of dimension-consistent design on other platforms, and demonstrations of the advantageous performance of EfficientViT on other hardware and compilers (Nvidia A100 with TensorRT, iPhone CPU with CoreML, and Android device with NNAPI).
> We hope our responses have addressed your concerns.
>
> Best,
>
> Authors

---

### Meta-Review · Area_Chair_hjyC · 2022-08-27

**Recommendation:** Accept
**Confidence:** Certain

**Metareview:**

This work proposes a purely transformer-based vision model for mobile vision purposes.

This proposition is somewhat surprising, since transformers did not excel at low-latency inference on resource-constrained hardware, especially compared to convolutional networks.

This is achieved by using a clever design that allows for reshape operations without the actual need of copying data as well as new techniques for latency-optimized network pruning.

While the methods itself are very technical and engineering-oriented, the overall result: a purely transformer-based low-latency, high-quality vision network is of general interest and is worth being shared with the wider community. Therefore I propose this paper to be accepted for NeurIPS 2022.



**Award:**

No

---

### Decision · Program_Chairs · 2022-09-14

Accept